# Innate Immune Response to Dengue Virus: Toll-like Receptors and Antiviral Response

**DOI:** 10.3390/v14050992

**Published:** 2022-05-07

**Authors:** Caroline Fernandes-Santos, Elzinandes Leal de Azeredo

**Affiliations:** Viral Immunology Laboratory, Instituto Oswaldo Cruz—Fiocruz, Rio de Janeiro 21040-360, Brazil; carol.uned@gmail.com

**Keywords:** Toll-like receptors, dengue virus, innate immunity, antiviral response, cytokines, interferons

## Abstract

Dengue is a mosquito-borne viral disease caused by the dengue virus (DENV1-4). The clinical manifestations range from asymptomatic to life-threatening dengue hemorrhagic fever (DHF) and/or Dengue Shock Syndrome (DSS). Viral and host factors are related to the clinical outcome of dengue, although the disease pathogenesis remains uncertain. The innate antiviral response to DENV is implemented by a variety of immune cells and inflammatory mediators. Blood monocytes, dendritic cells (DCs) and tissue macrophages are the main target cells of DENV infection. These cells recognize pathogen-associated molecular patterns (PAMPs) through pattern recognition receptors (PRRs). Pathogen recognition is a critical step in eliciting the innate immune response. Toll-like receptors (TLRs) are responsible for the innate recognition of pathogens and represent an essential component of the innate and adaptive immune response. Ten different TLRs are described in humans, which are expressed in many different immune cells. The engagement of TLRs with viral PAMPs triggers downstream signaling pathways leading to the production of inflammatory cytokines, interferons (IFNs) and other molecules essential for the prevention of viral replication. Here, we summarize the crucial TLRs’ roles in the antiviral innate immune response to DENV and their association with viral pathogenesis.

## 1. Introduction

Dengue virus (DENV) is mosquito-borne viral disease that has spread to tropical and subtropical regions which have great importance regarding global health. DENV is transmitted mainly by mosquitoes of the species *Aedes aegypti* and less frequently by *Aedes albopictus* [1,2]. Despite being an underreported disease, it is estimated that about 400 million new cases occur annually [3]. DENV belongs to the genus Flavivirus and the family Flaviviridae, and there are four distinct but closely related serotypes called DENV-1, DENV-2, DENV-3 and DENV-4 [4]. 

DENV infection causes a wide spectrum of clinical manifestations ranging from inapparent illness to severe and potentially fatal disease. In this context, the clinical manifestations of DENV infection can vary from asymptomatic forms to mild febrile forms (dengue fever—DF) and severe forms, accompanied by hemorrhagic episodes and increased vascular permeability, which can lead to shock and death (dengue hemorrhagic fever/Dengue Shock Syndrome—DHF/DSS) [5,6]. Based on a multicenter study called Dengue Control (DENCO), the World Health Organization (WHO) proposed a new classification for dengue cases in 2009. The WHO 2009 classification comprised three categories: dengue without warning signs (DwoWS), dengue with warning signs (DwWS) and severe dengue (SD). SD is characterized by severe bleeding, plasma leakage and/or organ impairment [7]. Aspects of the virus and host have been targeted in intensive research to understand the different clinical conditions presented by infected patients. 

Different hypotheses have been discussed, such as cytokine storm, antibody-dependent enhancement (ADE) and original antigenic sin, among others. Virulent strains and disorders of coagulation factors have also been correlated with severe disease. Despite this intensive research, no consensus has emerged regarding the exact immunopathogenic mechanisms involved, but these are likely to be multifactorial [8].

Pattern-recognition receptors (PRRs) recognize conserved microbial components termed pathogen-associated molecular patterns (PAMPs) or damage-associated molecular patterns (DAMPs). Toll-like receptors (TLRs) are the major PRRs that recognize viral PAMPs such as nucleic acids derived from viral genomes or generated during viral replication. TLRs’ recognition trigger cytokines and type I interferon (IFN) production, and therefore, they play crucial roles in innate and adaptive immune responses to viral infection [9].

The innate antiviral response to dengue virus (DENV) is implemented by a variety of immune cells and inflammatory mediators. If these components are in balance, viral clearance is efficient and, consequently, infections will be asymptomatic or will only have mild clinical manifestations. On the other hand, imbalance between these components can lead to the exacerbation of the immune response and greater disease severity. In this review, we summarize the crucial roles of TLRs during DENV infection and their impact on cell activation and cytokine, chemokine and IFN production. We present advances regarding the function of the TLR2 subfamily, TLR3, TLR4 and TLR7 and their association with disease severity. 

## 2. The Dengue Virus Life Cycle

The flaviviruses are small (50 nm in diameter) enveloped, spherical and icosahedral viruses. The viral genome is composed of a positive-sense RNA of approximately 11 kb that encodes three structural (capsid protein C, membrane protein M and envelope protein E) that are components of the virion and seven non-structural viral proteins (NS1, NS2A, NS2B, NS3, NS4A, NS4B and NS5) involved in viral replication and maturation [10]. 

DENV is inoculated by the mosquito vector into the skin epidermis where it encounters permissive skin-resident cells such as keratinocytes, fibroblasts, mast cells and immature dendritic cells (langerhans cells) [11]. These cells migrate to lymphoid organs, favoring its dissemination to peripheral blood mononuclear cells (PBMCs) and tissues. Human autopsy tissue studies detected DENV antigens in hepatocytes, kupffer cells, cardiac fibers, tissue endothelial cells, monocytes, lymphocytes and platelets, suggesting those cells are targets of DENV infection [12,13,14,15]. However, the mononuclear phagocyte lineages such as monocytes, macrophages and dendritic cells (DCs) are considered the primary targets of DENV infection [16]. These cells represent good cellular models of DENV infection in vitro [17], and they were described as the main target cells of DENV infection in vivo [18]. 

The replication cycle begins with the adsorption of the virus on the main target cells (monocytes, macrophages and DCs), through cell receptors expressed in the plasma membrane. Several different mammalian cell receptors have been proposed, including heparan sulfate [19], the mannose receptor (MR) [20], CD14 [21], heat shock protein 70 (Hsp70) and Hsp90 [22] and dendritic cell-specific intercellular adhesion molecule-3 grabbing non-integrin (DC-SIGN/CD209). DC-SIGN is one the best candidate receptors, being indispensable for DENV infection [23,24,25]. 

The virus enters permissive host cells via receptor-mediated endocytosis, similarly to other flaviviruses. After binding cell receptors, the virus is then endocytosed in a clathrin-mediated process. Once inside the cell, endocytic vesicle acidification leads to glycoprotein E rearrangement, fusion of the viral envelope with the endosomal membrane and release of the viral ssRNA into the cytoplasm. The viral ssRNA is then translated on the rough endoplasmic reticulum (RER) membrane, initially producing a polyprotein that is subsequently cleaved into individual proteins (structural and non-structural), while the viral ssRNA is replicated via a viral replication complex. Subsequently, the assembly of the immature virus occurs with the participation of protein E, PrM, C and the newly synthesized RNA, through a mechanism called budding. Finally, viral precursor membrane (prM) proteolysis occurs in the trans-Golgi network, mediated by the host enzyme furin, activating E protein homodimerization rearrangement and the formation of new, mature viral particles that are released from the host cells [26]. 

## 3. Immune Response to DENV 

### 3.1. Innate and Adaptive Immune Responses to Dengue Virus (DENV)

The immune system consists of cellular and molecular organization with specialized functions in the defense against foreign invaders and is fundamental to the maintenance of body homeostasis. [27]. The immune system has two central lines of host defenses, innate and adaptive, which cooperate to sense and eliminate pathogenic microbes [28]. Innate and adaptive immune responses are required in controlling DENV infection. As a mosquito takes a blood meal, DENV is introduced in the skin of susceptible hosts [29]. The tissue-resident macrophages of skin, the langerhans cells, are the first cells to be infected. They express PRRs that recognize viral PAMPs, and after PRR engagement, an antiviral response is trigged, initiating type I IFN production in the early stages of DENV infection. Furthermore, other inflammatory mediators are produced in response to infection, establishing an inflammatory microenvironment. Cytokines, chemokines and acute-phase proteins act in the direct destruction of pathogens and in signaling to effector cells [29,30].

Type I and III IFNs induce antiviral states in cells and aid in the production of proinflammatory and antiviral molecules through signaling pathways related to PRRs [31]. Type I IFN can be produced by all nucleated cells upon pathogen recognition. DCs and plasmocytic dendritic cells (pDCs) are specialized cells that secrete high amounts of type I IFNs during viral infections [32]; macrophages, monocytes and DCs produce type III IFNs, but epithelial cells are the main source [33].

Briefly, once activated, intracellular signaling pathways downstream of PRRs culminate in the activation of the transcription factors—IRF3 (Interferon Regulatory Factor), IRF7 and NF-κB, which direct the transcription and secretion of IFNs and inflammatory cytokines. Inflammatory cytokine production leads to the recruitment of leukocytes to the infection site. The IFNs bind to their receptors (IFNAR) and activate the Janus kinase (Jak)/Signal transducer and activator of transcription (STAT), leading to the induction of interferon stimulated genes (ISGs). ISGs exert many antiviral functions. The ISG products interfere with steps of the viral lifecycle, generating an antiviral state [31]. ISGs are regulated through IFNAR and PRR activation, but they can also be directly induced by IRF3 in an IFN-independent pathway [31,34].

A lot of evidence shows that type I IFNs would activate the antiviral response, accelerating viral clearance during DENV infection. We have suggested that the activation of the type I IFN pathway would be a marker of good prognosis during infection. We found the greater activation of pDC, main IFN producers, and higher plasma concentrations of IFN-α in mild cases of dengue [35]. Other studies demonstrated the robust production of IFN-α in mild cases [36,37]. In addition, genes related to the pathway or activation by type I IFNs such as STAT1, STAT3, IRF7 and IRF9 genes were upregulated in DF [38]. On the other hand, the decreased cellular expression of many ISGs (Mx1, Mx2, ISG15, Interferon-induced protein with tetratricopeptide repeats- IFIT2 and OAS3) were found in cases of DSS [39].

Studies in vitro demonstrated that type I interferons are antivirals for DENV, and this feature appears to be common to flaviviruses. Several ISGs, including viperin, Interferon-Induced Exonuclease (ISG20) and interferon-inducible transmembranes—IFITM2 and IFITM3—were identified as mediating resistance to DENV infection in vitro [40,41]. 

Given the importance of innate antiviral immunity, DENV have evolved several evasion strategies to avoid detection by the host innate immune system at their sites of replication. In this way, in vitro treatment with type I IFN-(α/β) before DENV infection protected human HepG2 cells from viral replication, but IFN treatment after DENV infection was not able to control viral replication, indicating that DENV evolved type I IFN antagonistic mechanisms [42]. Indeed, antagonistic mechanisms for the interferon pathway, which would result in the attenuation of type I IFNs production and/or their signaling, were described. The relationship between DENV infection and IFN effects was demonstrated by Muñoz-Jordán and colleagues (2003) [43]. The authors analyzed the ability of the 10 proteins encoded by DENV-2 to block the IFN system. For this proposal, they transfected A549 human cells with different plasmids that expressed each of the 10 nonstructural proteins encoded by the DENV2 genome [43]. Since the Newcastle disease virus (NDV) is highly sensitive to type I IFN [44], the potential DENV-encoded IFN antagonists were investigated based on their ability to facilitate NDV-GFP (green fluorescent protein) replication in transfected chicken embryo fibroblasts (CEFs). Three potential antagonists of IFN, NS2A, NS4A and NS4B, were identified, as they enhanced the viral replication of NDV-GFP. The authors demonstrated that NS2A, NS4A and NS4B proteins block IFN-induced signal transduction. Sometime later, they showed that NS4B was able to block STAT1phosphorylation [43,45].

Previous studies performed by Jones and colleagues (2005) showed that DENV infection reduces the cellular expression of STAT2 [46]. Another study by the same group demonstrated that the DENV NS5 protein is a potent and specific type I IFN antagonist. NS5 binds STAT2 and inhibits its phosphorylation, consequently inhibiting downstream events in the type I IFN response [47].

The initiation of dengue-virus-specific adaptive immune responses starts after the infection of immature dendritic cells (langerhans cells) in the skin. These cells undergo maturation and migrate to lymph nodes, activating TCD4+ and TCD8+ cells. Thus, after infection, the effector cells of innate and adaptive immunity release cytokines, amplifying the local inflammatory response [48]. 

During viral infection, it is expected that when glycoproteins (GPs) on the virus surface are recognized by the immune compounds, this leads to memory formation by triggering the adaptive immune system. Humoral and cellular response to these highly antigenic GP would provide an effective antiviral response [49]. In dengue, the viral serotypes share about 65–75% of their entire genome. The main targets of neutralizing antibodies are E, preM/M and NS1 DENV proteins. Domain III of the viral envelope is the main target of neutralizing antibodies, essential in the protection against infection [50]. Natural infection with any serotype results in long-lasting immunity to the homologous serotype (primary infection) but induces short-term immunity to the heterotypic serotype (secondary infection). Neutralizing antibodies are protective, while cross-reactive subneutralizing antibodies effectively participate in disease severity. Sometimes, the virus–antibody immunocomplex facilitates virus entry into immune cells bearing Fcγ receptor (FcγR), leading to higher viral load and consequently severe outcomes through a mechanism called antibody-dependent enhancement (ADE) [51]. So, instead of protecting, cross-reactive antibodies to the virus enhance the disease. The ADE is pertinent in the context of pre-existing immunity, and epidemiological studies revealed the association of severe dengue in individuals experiencing secondary infection [52,53]. 

In 1977, Halstead and O’Rourke pioneered studies on dengue and introduced the concept of ADE in DENV infection. PBMC from dengue-immune primates showed enhanced infection as compared with non-immune primates. Thus, it was hypothesized that virus–antibody immunocomplexes preferentially enter in primary monocytes and macrophages via the Fc receptor, increasing the phagocytosis and consequently viral loads [54,55,56]. 

The mechanism that DENV disposes to subvert immune response in humans does not to apply to animal models. Immunocompetent mice do not sustain a substantial infection, and this resistance is dependent on the IFN system [57]; however, ADE has been demonstrated in animal models [58,59] and in vitro models [60]. Despite that, few studies proved the ADE mechanism in humans. Recently, Waggoner and colleagues (2020) showed the association of sub-neutralizing antibody titers and DENV viral loads with disease severity in naturally infected patients, providing evidence for ADE in humans dengue cases [61].

Several studies suggested that CD4+ and CD8+ T lymphocytes are involved in the protection and resolution of DENV infection. DENV-specific human CD4+ and CD8+ T lymphocytes produce IFN-γ and lyse-infected target cells [62,63], suggesting that serotype-specific T lymphocytes are activated and functional in DENV infection [64].

In 2003, the phenomenon known as “original antigenic sin” was proposed to explain how specific DENV T lymphocytes could contribute to the severity of secondary infections [65]. According to “original antigenic sin”, during a secondary infection, the expansion of preexisting cross-reactive memory B and T cells dominate over memory cells activated by the infecting viral serotype [66]. These memory cells have low avidity for epitopes of the infecting serotype, being less effective in viral elimination. They display suboptimal degranulation and increased TNF-α and IFN-γ production [65].

### 3.2. Immunopathogenesis versus Protection

The “antibody-dependent facilitation of infection” and “original antigenic sin” are the main discussed phenomena associated with dengue severity. It is a consensus that cytokines, chemokines and other inflammatory mediators produced in an exacerbated way, the “cytokine storm”, by different cells of the immune system contribute to the changes in endothelial permeability observed in the severe dengue. Cytokine storm is associated with endothelial permeability, tissue damage and multiple organ failure [67].

During dengue infection, the host's innate immune response acts as a first-line defense in controlling viral replication before generating an adaptive immune response. The cells of innate immune system recognize pathogens through PRRs. TLRs are coupled to detect specific viral components and induce the production of IFNs and other pro-inflammatory cytokines. The effector action of these cells mainly results in an antiviral response, but the disbalanced responses could lead to disease pathogenesis. DENV is able to evade the host’s innate immune response, especially the type I responses. With the evasion of the host’s innate immune response, greater viral replication occurs in target cells, and consequently, innate immune cells are recruited and produce inflammatory mediators in an exacerbated manner, causing endothelial and multiple organ damage and dysfunction. The excessive production of inflammatory mediators generates an increase in the permeability of the vessels, which can lead to fluid leakage and severe manifestations of dengue [11].

## 4. Pattern Recognition Receptors

PRRs recognize PAMPs and/or DAMPs and provide an effective warning and trigger innate immune responses. Through PRRs, the immune system is able to recognize various classes of pathogens. This is only possible because they share small portions detected as “patterns” among each other. For example, a cell wall component characteristic of Gram-negative bacteria, lipopolysaccharide (LPS), is the target PAMP for mammalian host cells. LPS is recognized by TLR4, thus leading to the activation of the inflammatory response. PRRs are classified as receptors of innate immunity, which differentiates them from specific lymphocyte receptors (adaptive immunity). All innate leukocytes constitutively express these receptors on their cell surface, endosomes and/or cytoplasm [68]. TLRs are not only expressed in innate immune cells but also in nonimmune cells such as fibroblasts and epithelial cells. Recognition by PRRs triggers signaling cascades that induce the production and release of inflammatory cytokines, chemokines and IFNs. The innate response instructs the adaptive immune/specific response to the pathogen in question, thus inducing appropriate effector immune responses [69].

There are five main PRR superfamilies: Toll-like receptors (TLRs), NOD-like receptors (NLRs), RIG-like receptors (RLRs), AIM2-like receptors (ALRs) and lectin-C-like receptors (CLRs) [70]. In this review, we focus on the role of TLRs during DENV infection.

## 5. Toll-like Receptor Signaling

TLRs were initially discovered in Drosophila and were described to be essential for embryonic development and immunity. TLRs are a superfamily of receptors containing 13 members that have been described in mammals, classified as TLR1 to TLR13. Up to the present time, there are 10 different TLRs identified in humans. The TLR superfamily has also been subclassified into five subfamilies according to their positions on the phylogenetic tree: subfamily TLR1, subfamily TLR3, subfamily TLR4, subfamily TLR5 and subfamily TLR7. They are transmembrane receptors characterized according to their occurrence in the extracellular domain (leucine-rich domain) or the cytoplasmic domain (homologous to the IL-1 receptor (IL-1R) and also known as the toll IL-1R (TIR) domain). The TIR domain is responsible for interacting with adapter proteins, transmitting the signals from the TLRs and initiating signaling cascades that will culminate in the production of immunological mediators. These processes drive an antimicrobial effect [71]. 

The TIR domain can interact with five different adapters: myeloid differentiation primary-response gene 88 (MYD88) [72], MyD88-adaptor-like protein (MAL) [73], TIR-domain-containing adaptor protein inducing interferon-β (IFNβ) (TRIF) [74], TRIF-related adaptor molecule (TRAM) [75] and sterile α- and armadillo-motif-containing protein (SARM) [76]. MYD88 is used by the majority of TLRs, excluding TLR3, which uses TRIF and alternative TLR4 activation that can utilize MYD88, MAL, TRIF and TRAM [77]. 

MYD88, MAL, TRIF and TRAM form complexes with TLRs via TIR-TIR interactions, initiating pro-inflammatory responses, whereas SARM is known as an inhibitory adapter protein and has been described as a negative regulator of TLR signaling. The SARM inhibitory effect occurs through interaction between the BB-loop (highly conserved sequence in the TIR domain) and the other TIR-adapters blocking proinflammatory and antiviral effects [78]. MYD88 signaling activates tumor-necrosis-factor-receptor-associated factor 6 (TRAF6), IRF 1/5/7 and NF-κB, leading to the rapid production and release of TNF and type I IFN. MAL, TRIF and TRAM signaling activate TRAF6 and also TRAF3, culminating in NF-κB activation and IRF3/7 activation [77].

TLR4 was the first to have its location (cell surface) and ligand (LPS) defined by Poltorak et al. in 1998 [79]. Like TLR4, other TLRs such as TLR2 and TLR6 are located on the cell surface, while TLR7, TLR8 and TLR9 are located on endosomal membranes. TLR3 is typically located on endosomal membranes but can be also found on cell surfaces during viral infection [80,81]. Classically, TLRs expressed on the plasma membrane are associated with the recognition of PAMPs that are present on the surface of pathogens. These may or may not be soluble and are activated during the course of fungal, viral and bacterial infections. Endosomal TLRs recognize nucleic acids, which may be ssRNA, dsRNA or DNA, and are essentially associated with viral infections [82].

Studies to determine the function and characterization of TLR superfamily have advanced rapidly in the last two decades. It has been demonstrated that there is a certain promiscuity of recognition among TLRs and classification regarding the specificities of the pathogen for each one [83]. In relation to viral infections, endosomal TLRs were initially considered the most important with regard to recognition and response to infections, but recently, it has been shown that plasma membrane TLRs also play an important role in cell activation and response to viral infections. TLR4 activation, for example, appears to be essential in combating respiratory syncytial virus (RSV), vesicular stomatitis virus (VSV) and Ebola virus (EBOV) through recognizing viral glycoproteins and the subsequent production and release of type I IFN and proinflammatory cytokines via MYD88 and TLR4-TRAM-TRIF branches. Reviewed in ref [84]. From the perspective of DENV infection, new studies are being proposed year by year to discuss the role of TLRs both in the direct infection of cells and in the course of dengue infection. 

Depending on the TLR subfamily, some cell types stand out in response to DENV infection. The TLR1 subfamily mainly activates monocytes and dendritic cells and has also been associated with coagulopathies and pro-inflammatory imbalance [85,86,87,88,89,90,91]. TLR4 subfamily has been associated with vascular disorders such as increased vascular permeability and bleeding; the main pathways responsible for those events might be platelet and endothelial cell activation [85,86,92,93,94,95,96,97]. The TLR3 and TLR7 subfamily are associated with protective antiviral responses through NK, MC, DC and pDC cells [35,98,99,100,101,102,103,104,105,106,107] (Figure 1).

### 5.1. TLR1 Subfamily

The TLR1 subfamily contains the largest number of members: TLR1, TLR2, TLR6 and TLR10 [82]. To perform their functional role, the members of this subfamily act in association, a heterodimer between TLR2 and another family member. It is important to note that this association occurs between TLR2 and the other members, each of which is responsible for the recognition of different PAMPs independently [108]. TLRs induce intrinsic signaling pathways through the recruitment of specific adapter molecules, which leads to the activation of transcription factors such as NF-κB and IRFs that are crucial to triggering innate immune responses. 

The TLR1 subfamily is located on the cell membranes and predominantly recognizes bacterial components; therefore, TLR1 subfamily members are considered non-viral TLRs. However, there is growing body of evidence illustrating the importance of the TLR1 subfamily during viral infection, especially during dengue. In 2010, we reported that CD14 (+) monocytes expressing TLR2 and TLR4 were increased in peripheral blood from mild dengue patients compared to severe ones. The increased expression of CD16, intercellular adhesion molecule (ICAM) and TLRs suggested monocyte activation during infection. The in vitro infection of monocytes with DENV-2 (16681 strain) demonstrated a slight increase in TLR2 expression on monocytes after 24h. Interestingly, monocytes expressing increased levels of TLR2 had lower infection rates on day 2, and few monocytes co-expressing both TLR2 and DENV antigens were detected, suggesting that DENV may stimulate TLR2 expression through soluble factors or that the virus underwent clearance before TLR2 activation [85]. 

Subsequent studies found the up-regulation of TLR2 in DCs and pDCs of DHF patients but not in DF. It was proposed that the overexpression of TLR2 in DCs and pDCs could contribute to severe forms of dengue [109]. Corroborating these findings, a study carried out by De Kruif et al. (2008) analyzed TLR gene-expression profiling in 56 children with severe dengue from Indonesia. The authors showed the increased expression of TLR1, TLR2 and TLR4 transcript variant 4 (TLR4R4), while the TLR4 transcript variant 3 (TLR3R3) and TLR7 were decreased [110]. 

The study conducted by Chen et al. (2015) evaluated the expression of TLRs in DENV-2 infected PBMCs in vitro and, likewise, they observed an increase in TLR2 and TLR6 expressions in addition to higher IL-6 and TNF-α production. To determine which viral protein interacted with TLR2/6, PBMCs were treated separately with structural proteins C, E and pre-M, and with non-structural NS1, NS2A, NS2B, NS3, NS4A, NS4B and NS5. The results showed that NS1 induced cellular activation, IL-6 and TNF-α production via TLR2/6. Furthermore, signaling pathways of TLR2 and TLR6 were active and could contribute to the production of pro-inflammatory cytokines, since DV NS1 that stimulated TNF-α secretion was significantly reduced when both TLR2 and TLR6 were blocked. Importantly, the knockout (KO) of TLR6 increased the survival of DV NS1-treated mice, suggesting TLR6’s involvement with disease severity [111].

Monocyte populations are classified into three subsets: classical monocytes (CMs) CD14++CD16-, intermediate monocytes (IMs) CD14++CD16+ and non-classical monocytes (NMs) CD14+CD16++. CMs are rapidly recruited to the site of infection and are able to give rise to IM and NM during the infectious process [109,110]. It is known that DENV is able to invade and replicate in monocytes, macrophages and DC in the early stages of infection [17,112]; Aguilar-Briseño et al. (2020) inquired about the monocytes subsets and the expression of TLR2 in healthy donors and patients infected by DENV. The results evidenced that DENV infection causes alterations in the proportion of each subset of monocyte compared with healthy individuals. CMs were found diminished, while IMs and NMs were increased in DENV-infected patients. The authors highlighted that severe patients showed significantly higher TLR2 expression in CM monocytes as compared to mild ones. More importantly, the TLR2 recognition of the four DENV serotypes was evaluated in monocytes, and it was found that all DENV serotypes were able to activate the NF-κB in a TLR2-dependent manner [87]. Concerning to endothelial integrity and vascular response, the authors demonstrated that TLR2 blockage was able to prevent endothelial disruption induced by DENV2 infection. These results clarify the relation between monocytes and TLR2 expression in addition to supporting the idea that the activation of this receptor may be associated with severe symptoms such as plasma leakage [87].

Endothelial disorders can lead to hemorrhagic manifestations such as epistaxis, hematemesis and melena, with are frequently observed during dengue infection and thrombocytopenia is an important indicator of dengue severity [88]. Platelets are cell fragments derived from megakaryocytes and are susceptible to infection by DENV. Although capable of replicating the viral genome, they are unable to support the assembly of virions, a process known as abortive infection. Reviewed in ref. [92]. Human platelets express Syk-coupled C-type lectin receptors- CLRs (CLEC2 and CLEC5A) that can be activated and were reported as PRR for DENV [92]. Indeed, Sung et al. (2019) demonstrated that the activation of platelets via CLEC2 leads to degranulation, which has an impact on the activation of neutrophils via CLEC5A/TLR2 and culminates in the extrusion of extracellular trap neutrophils (NETs), showing an important association between platelets, NETs, severe hemorrhagic manifestations and lethality [89]. The study not only demonstrated the importance of cross-talk between platelets and neutrophilic activation, but also shed light on the co-participation between different PRRs (CLRs and TLR2) in the activation of the innate immune response to dengue and the clinical outcomes.

Several crucial points regarding the role of the TLR1 subfamily in signaling pathways and viral clearance during DENV infection have emerged. As described below, although the evaluation of infection using animal models is still controversial, it has contributed towards understanding issues that can be difficult to study in humans. Studies have investigated the role of receptors of the TLR1 subfamily through infection of DCs with DENV and antibodies responses evaluation. After all, DCs are the link between innate and adaptive immunity, and therefore, they are essential for the formation of immune memory [90,113].

It has been previously suggested that ADE in dengue infection is regulated by TLR2/MyD88 molecules. George et al. (2017) assessed the role of murine DCs derived from BM cells (BMDCs) from C57BL/6 (BL/6, H-2b) mice infected by DENV2 (strain DJB2001). Using BMDC from mice lacking TLR2, TLR3, TLR4 or TLR9, it was found that only TLR2-deficient BMDCs showed abolished IL-6 and TNF-α production in response to DENV2 infection. In addition, the role of heterodimeric complexes of TLR2 during the infection of BMDCs demonstrate that, in KO cells for TLR2, the infection was drastically reduced compared with wildtype (WT) cells, suggesting that TLR2 facilitates DENV infection [91]. 

The interaction between TLRs and their ligands depends on access by the ligands to these receptors, and can affect the signaling cascade [114]. George and colleagues also evaluated whether there was activation and maturation of these cells through TLRs, thereby resulting in the increased expression of Major Histocompatibility Complex (MHC) and co-stimulatory molecules during antigenic presentation for T cells. As expected, the experiments demonstrated that TLR2 and MYD88 are indeed necessary for the maturation of BMDCs during the antigen presentation process, measured by decreases in the expression of CD40, CD80/86, and MHC I/II in TLR2/MYD88-deficient mice. Interestingly, polarization to the Th2 -biased antibody response was also seen in a TLR2/MYD88-dependent manner, and the capability of this axis to cause ADE in subsequent heterotypic infections was shown [91]. 

Human studies have shown increased TLR2 expression during DENV infection [85,87,109,110]. It has already been seen that the activation of DCs by TLR2 produces a profile aimed at the production of antigen-specific antibodies and a Th2 response. It is important to emphasize that the Th2 profile usually appears in severe patients [56,115]. Thus, the evaluation of the cytokine production profile and Th2 polarization confirmed that DENV2 can lead to the production of the Th2 profile in a TLR2/MYD88-dependent manner. In fact, in assessing the occurrence of ADE during the secondary response in all animal groups, the rate of generation of neutralizing antibodies reached a maximum of 50%. However, the comparison between serum samples from animals that were KO for TLR2/MYD88 showed that there was a lower chance of ADE events than in samples from WT animals. Together, these results suggested that the activation of DCs via TLR2/MYD88 can lead to a shift in the Th1/Th2 response, frequently found in DHF patients, and might also be related to ADE events in subsequent infections [91].

### 5.2. TLR3 Subfamily

This subfamily, whose only member is TLR3 itself, differs from all the others, starting with the genomic segment that encodes it. While all other TLRs have one or two exons, TLR3 has five exons. Another particularity of this receptor is the fact that it recognizes the double strand of RNA produced during the replication cycle of several viruses, thus resulting in the production of large amounts of type I IFN [82]. Regarding the signaling trigger, TLR3 exclusively uses the TRIF adapter protein for NF-κB and IRF3 activation and the consequent production of type I IFNs and inflammatory mediators [77,84].

The relationship between TLR3 and DENV (Figure 2) was first demonstrated by Tsai et al. in 2009 [100]. The authors sought to elucidate the role of TLRs during DENV2 infection (NGC strain), using in vitro infection models of U937, THP-1 and HEK293 cells. Knowing that TLR3 is a receptor present in cell endosomes, endosome acidification is needed for viral recognition. For this purpose, cells were pretreated with chloroquine and/or bafilomycin A1 (which is known to prevent endosomal acidification). The results demonstrated that U937 and THP-1 cells required endosomal acidification for the production of IL-8 and IL-6. The authors next infected the HEK293-TLR3 cells and obtained similar results regarding the production of IL-8, which suggested that TLR3 has an important role in IL-8 production after DENV recognition. Additionally, TLR3 expression had a protective effect on the culture through decreasing the viral yield by up to 40-fold. Lastly, the authors also demonstrated the existence of colocalization of TLR3, TLR7 and TLR8 in the endosome, and suggesting that TLR7 and TLR8 could become activated, generating antiviral responses [100].

In another study by Liang et al. (2011), the TLR3 antiviral effects during DENV infection were confirmed. After confirmation of the constitutive expression of TLR1 to TLR8 (except TLR2) in HepG2 cells, it was shown that pretreatment with poly (I:C) (the main TLR3 agonist) was protective against DENV2 infection (New Guinea C strain), through reducing drastically the viral mRNA copies, and in addition to favoring the production of IFN types I and III [99]. Together, Tsai and Liang began to map out the essential role of TLR3 regarding infection control and cytokine and IFN production. Thus, it was hypothesized that TLR3 is essential for the proinflammatory response induction and control of DENV infection in different in vitro models, hence possibly indicating that it has a protective role during infection in vivo.

Understanding the susceptibility of skin cells, especially fibroblasts, to DENV infection is important for determining how viral replication occurs at the inoculation site. It is interesting to note that fibroblasts are susceptible to infection not only in vivo but also in situ. With regard to PRRs in this case, it has been suggested in the literature that TLR3 and Retinoic acid-inducible gene I (RIG-I)-like receptors (RIG-I) are the main PRRs responsible for of innate response and antiviral state in DENV infected skin fibroblasts. These two PRRs appear to work together during DENV infection, given their kinetic pattern of expression. During DENV2 infection, the peak expression of TLR3 was found to occur within the first 12 h p.i., whereas RIG-I expression increased from 24 h p.i. onwards [102]. Regarding keratinocytes, increased TLR3 expression through increased levels of transcriptional mRNA suggested antiviral type I IFN production and control of DENV replication [116].

Interestingly, DENV infection in the absence of RIG-I and melanoma differentiation-associated gene 5 (MDA5) increases the permissiveness of cells to the replication and propagation of the virus; however, it increases the production of IFN-β including through the increased expression of genes stimulated or related to IFN, OAS2, ISG15 and ISG56, as well as activates IRF3. In the absence of TLR3, there is an increase in virus replication with a decrease in the IFN-β production. This process is even more pronounced when in the absence of RIG-I, MDA5 and TLR3 [117]. 

It was demonstrated that TLR3 but not TLR7 or TLR8 was involved in the DENV-induced production of type III IFN (IFN-λ) in human DCs. Among NS proteins, NS1 of DENV was able to induce IFN-λ production in an NF-κB- and IRF-3-dependent manner. The study highlighted the IFN-λ-mediated immunologic effects during DENV infection. The KO of IFN-λR1 impaired DENV-induced DC migration through CCR7, the receptor for CCL19 and CCL21 chemoattractant cytokines, one prerequisite for the subsequent activation and shaping of adaptive immunity [118].

PRR signaling pathways culminate in the inflammatory and antiviral response, which together orchestrate the early innate host responses to DENV infection. Faced with the dichotomy presented above, it is clear that the different members of the PRR pathway can be regulated during infection by DENV. As mentioned, DENV has developed strategies to subvert the early host defenses by interfering with molecules involved in anti-viral responses. Indeed, DENV is able to inhibit the production of type I IFN through the negative regulation of TRAF6, a molecule involved in signaling by TLR3/7/8 and RIG-I/MDA-5. This inhibitory effect can occur through miRNA-146a and results in higher rates of viral replication [119]. TLR3 and RIG-I/MDA-5 are prominent targets for such downregulation, as these molecules act synergistically to produce the antiviral state against DENV1, so as to restrict the infection [117]. 

Even though DENV has mechanisms for escaping from the immune system through interfering with the PRR signaling cascades in some cell types, the virus is unable to manipulate all cells or all cells at the same time. Thus, immune cells are able to produce antiviral molecules and consequently the viral clearance is achieved. It is not uncommon for these inflammatory mediators to be overproduced, which leads to cytokine storm, largely related to severe forms of the disease [120]. Our group already demonstrated increased levels of inflammatory mediators such as nitric oxide (NO), TNF-α and IL-8 in DENV- infected monocytes in vitro directly linked with P2X7r activities. P2X7r is responsible for the inflammasome leucine-rich repeat (LRR)-containing proteins (NLRP3) activation cascade, resulting in caspase-1 activation and proinflammatory cytokines production [106]. Indeed, TLR3 and the inflammasome NLRP3 activation were associated with the pathogenesis of dengue, especially in the development of hemorrhages [121].

In order to clarify whether other mechanisms are involved in occurrences of hemorrhagic manifestations, the endothelial dysfunctions should be recognized. One of the main agents responsible for altered vascular permeability is the nitric oxide synthase family; in particular, the inducible isoform that is released during inflammatory responses (iNOS). iNOS induces the production of NO, an unpaired-electron free radical that is capable of acting directly on pathogens and which leads to cellular metabolic stress [122]. Cheng et al. (2015) demonstrated that TLR3 is required for iNOS/NO synthesis through NF-κB activation. The direct inhibition of TLR3 decreased NF-κB activation and production of nitrite and iNOS in DENV2 infected RAW264.7 cells indicating that TLR3 regulates iNOS/NO production during host antiviral responses [107]. 

The immune response is complex and composed of interconnection between different pathways, receptors and ligands. Another class of receptors that is influenced by the action of TLR3 during DENV infection is notch receptors (notch 1-4) and their ligands (Dll1-4) [123]. These receptors are transmembrane proteins located on cell surfaces and are involved in a variety of processes such as embryonic development and immune cell communication. For example, they have an essential role during antigenic presentation between APCs and lymphocytes [124,125]. Interaction between notch receptors and their ligands results in the cleavage and release of the intracellular portion. After the release, translocation to the cell nucleus occurs, where it promotes the differential expression of several genes involved in the host immune responses [126]. The investigation of Dll1 and Dll4 levels during DENV infection showed that TLR3 knockdown of monocyte-derived macrophage cells produced about 80% less ligands. In addition, KO cells showed the decreased production of IFN-β, confirming previous results regarding the antiviral role of TLR3. These results suggest that DENV infection upregulates Dll1/4 expression relating to IFN-β production [127].

Classically unrelated effects from TLR3 during DENV infection have been shown, such as its role in microglial migration, through the dsRNA recognition pathway during infection [128]. Vitamin D production and supplementation can also influence the course of the DENV infection in monocyte-derived dendritic cells (MoDDCs). It has been demonstrated that vitamin D supplementation reduced the percentage of DENV positive cells and was associated with downregulation of TLR3 expression and also altered the production of IL-8, IL-12 and IL-10 [98].

Regarding the activation of natural killer (NK) cells during infection, we demonstrated that mild patients who were naturally infected by DENV had increased frequencies of CD107a (a degranulation marker), TLR3 and tumor necrosis factor- (TNF-) related apoptosis inducing (TRAIL). The type I IFN was involved in TRAIL expression on NK cells during DENV-2 stimulation in vitro [35]. The production of type I IFN would enhance the NK cell cytotoxicity, leading to efficient viral clearance. Lastly, corroborating the relationship of TLR3 with mild outcomes, the increased expression of TLR3 in DCs of patients with dengue fever (DF) was described (106). Although the exact mechanisms for how TLR3 affects DENV replication remain unclear, a lot of evidence suggested that TLR3 is a key molecule for DENV-protective innate immune responses.

### 5.3. TLR4 Subfamily

TLR4 is the only member of this subfamily. It was the first mammalian TLR to be discovered and has now been correlated with the recognition of LPS. In addition to microbial components, TLR4 also recognizes some endogenous components, such as HSPs (HSP60 and HSP70), for example [82]. The formation of this receptor occurs through homodimerization, and its signaling occurs classically through MyD88 or through an alternative route, using the toll-interleukin-1 receptor (TIR) domain-containing adaptor protein (TIRAP) molecule. Another particularity of TLR4 is its association with CD14 for the recognition of proteins with irregular characteristics, such as rough LPS. This recognition, in association with CD14, leads to the activation of IRF3 via TRIF [77].

During DENV infection, an association between TLR4 and severe dengue has already been seen. Kruif et al. (2008) demonstrated that some transcribed variants of TLR4 undergo an increase (TLR4R3) or decrease (TLR4R4 and TLR4 cofactor CD14) in children with the severe form of the disease [110]. We have shown the increased expression of TLR4 in PBMCs from infected patients during the febrile phase [85]. Although we and others have shown the differential expression of TLR4 in naturally infected patients, the way in which this receptor influences the course of DENV infection has not yet been elucidated. Regarding TLR4 roles, we can revisit the study of Modhiran et al. (2015), who, following findings with regard to other viruses, investigated the role of NS1 protein during DENV infection and its relationship with TLR2 and TLR4. Using an in vitro and a mouse model of DENV infection, they demonstrated that NS1 activates leukocytes and leads to proinflammatory cytokine production [86]. An association between the recognition of NS1 by TLR4 and hemorrhagic manifestations was also seen and, according to authors, NS1 protein might contribute to fluid leakage and imitates what would happen during septic shock, certainly with regard to the co-location between NS1s and TLR4 observed in cell membranes. Furthermore, when NS1 blockers were used, the hemorrhagic manifestations ceased, thus endorsing what they hypothesized. Cell activation via TLR4 led to the production of the inflammatory mediators TNF-α, IL-6, IFN-β, IL-1β and IL-12, as seen with other TLRs, and could lead to greater disease severity [86].

Beatty et al. (2015) published a trial in which mice were challenged with a sublethal dose of NS1 from all four serotypes and a subsequent challenge with a lethal dose of NS1 of the DENV2 serotype. The first challenge gave rise to a protective effect in relation to the second. However, NS1 of these four serotypes is potentially pathogenic, thereby leading to hemorrhagic manifestations and endothelial cell dysfunction in cultures of a human pulmonary microvascular cell (HPMEC) in vitro [93]. In a subsequent study, the same group demonstrated the effects of NS1 on the human lung microvascular endothelial cell line (HPMEC-ST1.6R), in which the activation of TLR4 led to sialic acid dysfunction in the glycocalyx. When treated with *Rhodobacter sphaeroides* lipopolysaccharide (LPS-RS), a TLR4 antagonist, sialic acid dysfunction was diminished, thus indicating a possible relationship between disease severity and TLR4 activation [94].

The relationship between NS1 and the activation of PBMCs has been assessed in several subsequent studies. These studies have demonstrated that under NS1 antigen interaction, these cells respond with the robust production of proinflammatory cytokines and chemokines. The importance of the correct purification of commercialized NS1 for microbial products has also been highlighted, such that there is no interference in different tests [129,130].

Lastly, the activation of inflammatory response trigged during innate immunity is integrated and synchronized in order to achieved appropriate responses and that avoids immunopathology. In this regard, the role of TLRs in other blood elements contributing to DENV pathogenesis needs to be clarified. As mentioned, in relation to hemorrhagic manifestations frequently observed in cases of dengue, the blood components responsible for the clotting process, i.e., platelets, cannot be ignored. Quirino-Teixeira et al. (2020) studied the mechanisms of platelet activation through TLRs. The NS1 protein leads to the activation and degranulation of platelets. However, differently to what was observed in PBMCs, TLR activation was not via the TLR2/6 heterodimer but, rather, was dependent on TLR4 activation [95]. Indeed, it has been shown that, in recognizing LPS via MYD88, this receptor leads to platelet activation and aggregation [96]. Isolated platelets incubated with a DENV supernatant containing NS1 expressed activation markers (P-selectin) and apoptosis markers (phosphatidylserine—PS) on their surface, indicating that NS1 leads to platelet activation via TLR4 signal transduction. The NS1 protein and the host's TLR4 have also been shown to play crucial roles in hemorrhagic manifestations, through influencing the duration and intensity of bleeding [97]. It appears that platelets start producing inflammatory mediators when there is transcription of the viral genome (serving as PAMP and being recognized by platelets), and both the pro-aggregation activation and production of cytokines are augmented through the recognition of NS1 by TLR4 [95].

### 5.4. TLR7 Subfamily

TLR7, TLR8 and TLR9 are the components of the TLR7 subfamily, located inside the endosome, as is TLR3. These receptors have been described as viral detection receptors and recognize RNAss (TLR7/8) and CpG DNA (TLR9), thus sequentially activating a signaling cascade with antiviral effects through the production of IFNs and proinflammatory via MYD88 [82,84].

One of the first studies illustrating the role of TLR7 during DENV infections demonstrated that the recognition of ssRNA by TLR7 depends on the exposure of the viral RNA in the endosome. Furthermore, increased antiviral response was demonstrated through the production of type I IFN by pDCs, compared with other cells; TLR7 the activation also led to the production of IL-8. The activation level of TLR7 and the cytokine production were highly dependent on the structure of the viral RNA molecule [104].

Sun et al. (2009) demonstrated that DENV-infected pDCs produced type I IFN and induced TLR7 expression indicating the involvement of TLR7-dependent signaling pathways in DENV recognition [105]. Comparably, Décembre et al. (2014) demonstrated that after the depletion of pDCs from the total PBMCs and the co-culturing of these PBMCs with DENV-infected cells, the production of IFN-α in response to DENV infection could no longer be detected; On the other hand, in co-culture models using pDCs, infected cells transmitted the virus to the pDCs, thus leading to the activation, cell maturation and establishment of antiviral status in TLR7 dependent manner; viral proteins alone did not signal massive type I IFN production by pDCs, and the effect was dependent on the viral RNA recognition by TLR7 [131].

The activation of TLR7 does not occur in the same way at subsequent infections. In Thai patients who were naturally infected by DENV, the expression and functionality of TLR7 were suppressed during secondary infections, along with those of TLR3 and TLR4.In vitro studies indicated that infection with DENV in presence of antibody complexes reduced TLR expression suggesting that ADE may suppresses TLR-dependent signaling pathways via Immunoglobulin Fc receptors (FcγRI and FcγRIIa) ligation [132]. Another study correlated clinical severity with adiposity levels in children. It was seen that a higher weight/age ratio in infants was a risk factor for DHF. In cells of myeloid origin from children with good nutritional status, TLR7 and TLR8 activated TNF-α production more robustly [133]. 

A study carried out among naturally DENV-infected Brazilian patients demonstrated that it was possible to identify seven genes with differential expression between DF and DHF: MYD88, TLR7, TLR3, MDA5, IRF3, IFN-α and CLEC5A. Among these genes, the two most important ones were listed as MYD88 and TLR7, since they were associated with DHF patients. However, this finding did not negate the importance of these particular genes and the other genes (TLR3, MDA5, IRF3, IFN-α and CLEC5A) for the establishment of an efficient antiviral response [134].

Knowing that DCs expressed TLR7 and TLR9, the one or combined TLR7 and TLR9 activation can occur leading to synergistic cytokine production. Interestingly, Lai et al. (2018) demonstrated that DENV infection activated TLR9 in human DCs. Knockout of TLR9 decreased innate immune response triggering and the viral loads increased. It has also been reported that DC activation occurs from the mitochondrial DNA release that occurs at infection, as a result of ROS production and the activation of the inflammasome, thus causing disturbances in the association between transcription factor A mitochondria (TFAM) and mtDNA, and consequently leading to the formation of pores on the mitochondrial membrane [135]. We have demonstrated that ROS production is also observed in monocytes infected by DENV, as a result of P2X7r activities [106]. Torres and colleagues observed that DF patients have a higher expression of TLR9, compared with DHF patients, throughout the course of the disease. The higher expression of TLR9 is associated with the production of type I IFN. It was demonstrated that type I IFN responses were impaired in PBMCs that had been stimulated with DENV and CpG, suggesting that the virus affected the IFN signaling pathway [109]. 

## 6. Conclusions

In this review, several studies demonstrating outcomes relating to different TLRs and subfamily characteristics were discussed. Studies strongly suggested that during the course of DENV infection, TLR2, TLR6 and TLR4 are associated with unfavorable outcomes for the host and being a potential therapeutic target. TLR3 was seen to have a crucial role in DENV clearance and TLR7, in type I IFN production, in manifold studies. These appeared to contribute to the establishment of efficient and protective antiviral innate immune defenses. Despite these major recent advances, TLR-mediated signaling during DENV infection is not fully understood. Besides that, there is no effective treatment or available vaccine for DENV infection. Knowledge of TLR-mediated signaling during DENV infection may help in the development of therapeutic procedures against specific viral targets as well as the development of TLR agonists that could be used in the vaccine design. PRRs are the first responder of the immune system. Knowledge about their biological features at different points of infection may provide pointers towards clinical outcomes and proper clinical treatments.

## Figures and Tables

**Figure 1 viruses-14-00992-f001:**
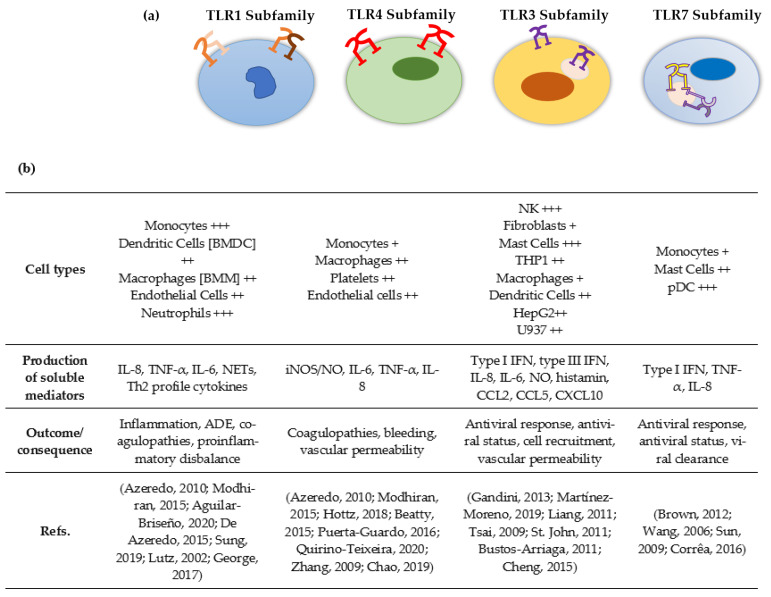
TLR response to DENV infection. Schematic drawing demonstrating the four different subfamilies of TLRs in different cells, production of soluble mediators and protective and/ or pathogenic outcomes. (**a**) Heterodimer and/or homodimer formation and its location on cell membrane and/or endosome. (**b**) Production of soluble mediators, expression in different cell types and protective and/or pathogenic outcomes of TLR subfamily activation during DENV infection [35,85,86,87,88,89,90,91,92,93,94,95,96,97,98,99,100,101,102,103,104,105,106,107]. +++ High expression, ++ intermediate expression, + low expression. Abbreviations: iNOS/NO—Inducible Nitric Oxide Synthase/Nitric Oxide; IL-6—Interleukin 6; IL-8—Interleukin 8; TNF-α—Tumor Necrosis Factor alpha; NETs—Neutrophil Extracellular Traps.

**Figure 2 viruses-14-00992-f002:**
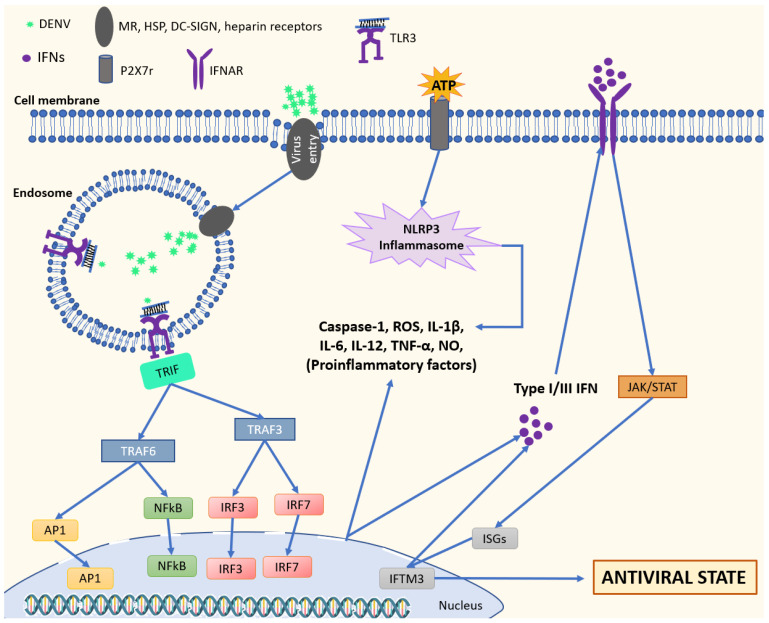
Antiviral state by activation of nucleic acid sensing by TLR3. TLR3 can recognize dsRNA as consequence of DENV infection. TLR3 activation stimulates production of proinflammatory factors and IFNs through transcription factors such as AP1, NFkB and IRF3/7. Activation of IFNAR by IFNs results in the transcription of hundreds of types IFN I and III and ISGs (via JAK/STAT) such as IFITM3 that promote IFN release in the endosomal compartment. During DENV infection, release of DAMPs and soluble mediators such as ATP and TNF-α are recognized by several cellular receptors such as P2X7r (ATP), triggering inflammasome assembly and activation. Consequently, capase-1, IL-1β, ROS and TFAM are produced and play roles in antiviral response and mitochondrial DNA releasement that may result in proinflammatory state and cytokine storm. Abbreviations: mannose receptor-MN, heat shock protein-HSP, dendritic cell-specific intercellu-lar adhesion molecule-3 grabbing non-integrin-DC-SIGN, interferon-induced transmembrane protein 3 -IFITM3, Interferon Stimulated Genes-ISGs, Tumor necrosis factor receptor-associated factor-TRAF, reactive oxygen species-ROS, interferon regulatory factor-IRF, TIR-domain-containing adaptor protein inducing interferon-β (IFNβ)-TRIF, transcription factor A mitochondria -TFAM.

## Data Availability

Not applicable.

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
