# Peer review of "Innate Immune Response to Dengue Virus: Toll-like Receptors and Antiviral Response"

_viruses, 2022, doi:10.3390/v14050992_

Round 1

Reviewer 1 Report

The authors present a review of the immunological response to dengue virus infection with a focus on Toll-Like Receptors. The review is extensive and appropriately organized to provide a logical flow. Unfortunately, the manuscript will need to be edited extensively for English grammar and sentence structure in order to improve the readers comprehension. I had a hard time deciphering the authors points in some sections. I started making suggestions in the attached but it was too extensive to do for the entire document. Figure 1 needs a better description and the Conclusions section can be expanded with specific examples of current knowledge gaps.

Reviewer 2 Report

The review article summarizes the role of TLRs in Dengue virus infection. The authors have tried to highlight the dual effects of the TLRs during the disease. However, there are many major issues that the authors should address. First of all, many sentences throughout the manuscript were challenging to understand. Secondly, many places throughout the manuscript miss references (not cited in the right places). It would be easier for readers to follow if references were cited immediately after the findings were discussed. I encourage authors to recheck the references thoroughly and cite them in the right places. Thirdly, the authors have decided the order of this paper. This review would be far better if the details follow the sequence: entry of DENV into the host, innate immune response, adaptive immune response, reinfection, severe disease conditions, and role of TLRs in every step of the disease pathogenesis. I recommend including at least one section/paragraph to couple TLRs with the Dengue infection life cycle.

Following are some places that I have spotted in the manuscript that are confusing. I hope that these comments will help the authors improve the review. The review itself is a good summary of the recent TLRs studies in Dengue virus infection and can be helpful to the field.

Abstract

  • Reorganizing it by focusing on DENV infection and TLRs will improve the abstract.
  • Line 8-9: the authors have stated interferons as the mechanism. (The process they induce is the mechanism, not the interferons)
  • Line 11: “….such as toll-like receptors (TLRs) the main point of this review” The sentence is not right.

Introduction

  • Line 28: “……..which have great importance regarding global 2) health”. It is not clear whether the authors are talking about Flaviviruses or the DENV specifically.
  • Line 32: The sentence “The viral…..are inserted” is not clear and correct.
  • Line 34: the authors have mentioned the amino-terminal portion of the genome which not correct. The genome does not have an amino-terminal portion.
  • Reference number 3 is not correct.

Immune response to DENV

  • Line number 75: there should be provided instead of provide
  • Line number 77: it should be “virus-antibody immunocomplex” instead of “immunocomplex virus-antibody”.
  • Line number 77: Facilitates instead of Facilitate
  • Line number 82: “…. Extend to those cells which…..” instead of “…. Extend to those cells who…..”
  • Line no 83: The sentence “Activation of….. dengue severity” is not understandable.
  • Line no 96: The sentence “ The interaction……….DENV infection” is repetitive.
  • Line no 105: “………happens with natural killer cells (NK) and natural killer T (NKT) cells” It is not understandable.
  • Line no 106: “It seemed” instead of “It was seemed”.
  • Line no 113: Please mention what types of cells produce Type I and II IFNs.
  • Line no 120: Please add references for the sentence “In vitro….flaviviruses”.
  • Line no 128: remove the word seem.
  • Line no 130: Please rewrite the sentence “In this study…..Sendai virus (SeV).
  • Line no 130: The sentence has mentioned the catalytic effects of NS4B protein, but has not explained
  • Line no 148: Please mention saliva of which animal.
  • Line no 150: The word “respectively” is unnecessary.
  • Line no. 151: I could not understand the sentence “alternations……DENV infection.”
  • Line no 155-157: rewrite the sentence.
  • Line no 158: Either such as or for example should be used. Please do not use both.
  • Line no 161: Sentence “Infection both …..immune system” is not correct
  • Line no 164: rewrite the sentence “This is the case…….(adaptive immunity).
  • Line no 180: No need to mention “smaller families”
  • Line no 195: What is the meaning of acting modulating signaling.
  • Line no 201: what is the meaning of “RNAm stability”
  • Line no 222: Add references.
  • Line no 227-242: In the figure legend of figure 1, please correct the English language. Red and black are mentioned in it but could not understand their meaning.
  • Line no 246: Directly write “a heterodimer” instead of writing “through forming a heterodimer”.
  • Line no 252-260: Rewrite this paragraph
  • Line no 263: What is the meaning of “we group provided”
  • Line no 264: The word ‘infected” is repeated.
  • Line no 277: HLA-DR molecules have not been explained before in this review.
  • Line no 284: please check the language of the sentence “ The …those molecules.”
  • Line no 303: Please check the language “the affirmation….DENV-2”
  • Line no 326: Could not understand “presented increased”
  • Line no 329-333: Rewrite the sentences.
  • Line no 354: What is the meaning of the sentence “It will see….in humans”
  • Line no 363: Could not understand “ it was decisive to primary “
  • Line no 379: could not understand the sentence “In parallel…BMDCs”
  • Figure no 2: Line no 426-436: Do not put references here. Please add the references in the explanation given in the main text. In the figure, the interaction of TNF-alpha with P2X7r is missing.
  • Line no 467: what is the meaning of “For example” in this sentence.
  • Line no 469: Add references.
  • A figure dedicated to TLR3 would be better.
  • Line no 512: the meaning of words seems and likely is the same.

Round 2

Reviewer 2 Report

I am glad that the authors have improved this review by responding to the weakness and revising it correspondingly. The revision is in good shape. I was trying to help and make it better and hope that the authors did not mind my comments last time.
